

# Mitogenomics of five *Olidiana* leafhoppers (Hemiptera: Cicadellidae: Coelidiinae) and their phylogenetic implications

Xianyi Wang, Jiajia Wang and Ren-Huai Dai

Institute of Entomology, Guizhou University, Guizhou Provincial Key Laboratory for Agricultural Pest Management of the Mountainous Region, Guiyang, Guizhou Provincial, China

## ABSTRACT

Similar morphological characteristics and limited molecular data of *Olidiana* resulted in their unknown phylogenetic statuses and equivocal relationships. To further understand the genus *Olidiana*, we sequenced and annotated five *Olidiana* complete mitochondrial genomes (mitogenomes). Our results show that *Olidiana* mitogenomes range from 15,205 bp to 15,993 bp in length and include 37 typical genes (13 protein-coding genes, 22 tRNAs, and 2 rRNAs) and a control region. Their nucleotide composition, codon usage, features of control region, and tRNA secondary structures are similar to other members of Cicadellidae. We constructed the phylogenetic tree of Cicadellidae using the maximum likelihood (ML) and Bayesian inference (BI) methods based on all valid mitogenome sequences. The most topological structure of the obtained phylogenetic tree is consistent. Our results support the monophyletic relationships among 10 subfamilies within Cicadellidae and confirm Iassinae and Coelidiinae to be sister groups with high approval ratings. Interestingly, *Olidiana* was inferred as a paraphyletic group with strong support via both ML and BI analyses. These complete mitogenomes of five *Olidiana* species could be useful in further studies for species diagnosis, evolution, and phylogeny research within Cicadellidae.

# INTRODUCTION

Cicadellidae is an extremely diverse family belonging to the order Hemiptera and contains an estimated 200,000 species (*Oman, Knight & Nielson, 1990*; *Dietrich, 2005*). Since the proposal of Cicadellidae as a paraphyletic group (*Cryan & Urban, 2012*; *Dietrich et al., 2017*), the phylogenetic relationships among its members have gained particular interest. Recently, mitochondrial genomes (mitogenomes) have been widely used to infer phylogenetic relationships among members of Cicadellidae. However, the primary focus has been on five subfamilies, namely, Cicadellinae, Deltocephalinae, Iassinae, Idiocerinae, and Typhlocybinae, and has included few representatives of Coelidiinae (*Du, Dai & Dietrich, 2017a*; *Du et al., 2017b*; *Song, Cai & Li, 2017*; *Song, Zhang & Zhao, 2019*; *Wang, Li & Dai, 2017*; *Wang et al., 2019b*; *Wang et al., 2019c*; *Yu et al., 2017*). Coelidiinae, one of the most diverse subfamilies within Cicadellidae, comprises 129 genera and more than 1,300 species (*Li & Fan, 2017*; *Nielson, 1980*; *Nielson, 2015*; *Viraktamath & Meshram, 2019*;

Corresponding author
Ren-Huai Dai, rhdai@gzu.edu.cn

*Wang et al., 2019a*). Previous studies that focused on morphological characteristics and molecular fragments (*H3*, *28S*, and *12S*) showed that Coelidiinae is closely related to Cicadellinae, Evacanthinae, and Typhlocybinae (*Dietrich & Deitz, 1993*; *Dietrich et al., 2001*; *Dietrich, 2010*). Furthermore, *Wang et al. (2019b)*, *Wang et al. (2019c)*, *Wang et al. (2020a)* and *Wang et al. (2020b)* reported that Coelidiinae and Iassinae form a clade at one branch and a sister group with Macropsinae. The phylogenetic relationships among members of Coelidiinae are inconsistent based on morphological characteristics and few published mitogenomes, and additional mitogenomes may provide a better understanding of the phylogenetic relationships among the genera and species of this subfamily and among members of Cicadellidae (*Dietrich et al., 2017*).

Within Coelidiinae, *Olidiana McKamey (2006)* is a relatively large genus, with 99 reported species from the Oriental and Palearctic realms (*Li & Fan, 2017*; *Nielson, 1982*; *Nielson, 2015*; *Viraktamath & Meshram, 2019*). Some species of *Olidiana* are relevant agricultural and forest pests and cause harm by directly feeding on plant sap or by indirectly spreading viral diseases (*Frazier, 1975*; *Li & Fan, 2017*; *Nielson, 1982*). *Olidiana* exhibits morphological characteristics extremely similar to those of other Coelidiinae genera, making species distinction a challenging task. Furthermore, some species of *Olidiana* have been incorrectly identified, resulting in more than one synonym for the same species (*Cai & Shen, 1998*; *Li & Fan, 2017*; *McKamey, 2006*; *Nielson, 1982*; *Nielson, 2015*; *Walker, 1851*; *Xu, 2000*; *Zhang, 1990*). In addition, the taxonomic statuses of some species of *Olidiana* are constantly changing, and several new genera (*Singillatus*, *Tumidorus*, and *Zhangolidia*) have been established by revising this genus (*Li & Fan, 2017*; *Nielson, 2015*). Collectively, generic classification remains unsatisfactory, making it challenging to determine phylogenetic relationships. Therefore, it is necessary to utilize the mitogenomes of *Olidiana* species to classify and determine the genetic relationships among Coelidiinae species.

At present, 116 partial or complete mitogenomes of Cicadellidae species have been deposited in GenBank. However, only three valid Coelidiinae mitogenomes [*Olidiana* sp., KY039119; *O. ritcheriina*, MK738125; and *Hiatusorus fascianus* (= *Taharana fasciana*, NC036015)] have been reported (*Wang, Li & Dai, 2017*; *Wang et al., 2019c*). In the present study, five *Olidiana* species, namely, *O. alata*, *O. longsticka*, *O. olbliquea*, *O. ritcheri*, and *O. tongmaiensis*, representing five main groups, were identified based on their morphological characteristics (*Li & Fan, 2017*; *Nielson, 1982*; *Nielson, 2015*). Their mitogenomes were sequenced and annotated, and the general characteristics of the mitogenome sequences were analyzed and compared. In addition, a phylogenetic tree was constructed using Bayesian inference (BI) and maximum likelihood (ML) methods to evaluate the relationships among Cicadellidae species.

## MATERIAL AND METHODS

### Sample collection and DNA extraction

Details of sample collection are presented in Table S1. All specimens were preserved in absolute ethanol and stored at −20 °C until analysis. Genomic DNA was extracted from

muscle tissues of adult males using the DNeasy® Tissue Kit (Qiagen, Germany). Total genomic DNA was eluted in 70 μL of double-distilled water. The remaining extraction steps were performed according to the manufacturer's protocol. The obtained genomic DNA was stored at −20 °C until further analysis.

## Mitogenome sequencing and assembly

Five *Olidiana* mitogenomes were sequenced using a next-generation sequencing platform (Illumina HiSeq 4000, Berry Genomic, Beijing, China; 6 GB raw data). Clean sequences were assembled using Geneious Primer version 2019.2.1 (*Kearse et al., 2012*), with *O. ritcheriina* (MK738125) (*Wang et al., 2019c*) as a reference.

## Mitogenome annotation and sequence analysis

The locations of 13 protein-coding genes (PCGs) were identified using the ORF Finder tool of the National Center for Biotechnology Information and the invertebrate mitochondrial genetic code. Uncommon start and stop codons were identified by comparing our sequences with those of other Cicadellidae species. The locations and secondary structures of 22 transfer RNA (tRNA) genes were determined using tRNAscan-SE (*Schattner, Brooks & Lowe, 2005*) and ARWEN version 1.2 (*Laslett & Canbäck, 2008*). Ribosomal RNA (rRNA) genes were identified based on the loci of adjacent tRNA genes and compared with those of other Cicadellidae species (*Wang, Li & Dai, 2017*; *Wang et al., 2019c*). Repeat sequences within the control region were determined using the Tandem Repeats Finder tool (http://tandem.bu.edu/trf/trf.submit.%20options.html) (*Benson, 1999*). The annotated mitogenome sequences of the five *Olidiana* species have been deposited in GenBank with the accession numbers MN780581–MN780585.

Base composition and relative synonymous codon usage (RSCU) in the mitogenomes were analyzed using MEGA version 6.06 (*Tamura et al., 2013*). Strand asymmetry was calculated using the following formulas: AT skew = $(A − T)/(A + T)$; GC skew = $(G − C)/(G + C)$ (*Perna & Kocher, 1995*). Intergenic spacers and overlapping regions between genes were manually counted.

## Sequence alignment and phylogenetic analysis

To determine the phylogenetic relationships among members of Cicadellidae, 74 species from 12 subfamilies of Cicadellidae as well as 6 treehopper species were included, with two Cercopoidea species (*Tettigades auropilosa* and *Cosmoscarta bispecularis*) used as outgroups (Table S2). Phylogenetic analysis was performed by independently aligning the sequences of 13 PCGs and 2 rRNA genes. For each PCG sequence, terminal codons were removed before alignment using MAFFT version 7.0 in the Translator X online server (http://translatorx.co.uk/) with the L-INS-i strategy (*Abascal, Zardoya & Telford, 2010*; *Castresana, 2000*). Each rRNA gene was individually aligned using MAFFT with the G-INS-I strategy, and poorly aligned sites were removed using Gblocks 0.91b (*Katoh, Rozewicki & Yamada, 2019*). The resulting 15 alignments were concatenated using MEGA version 6.

Five datasets were concatenated for phylogenetic analysis: (1) PCGs, all codon positions of the 13 PCGs with 10,044 nucleotides; (2) PCG12, first and second codon positions of

the 13 PCGs with 6,696 nucleotides; (3) AA, amino acid sequences of the 13 PCGs with 3,348 amino acids; (4) PCG12R, first and second codon positions of the 13 PCGs and 2 rRNA genes with 8,448 nucleotides; and (5) PCGR, all codon positions of the 13 PCGs and 2 rRNA genes with 11,796 nucleotides. The substitution saturation of four datasets (PCG, PCG12, PCG12R, and PCGR) was tested by plotting the number of transitions and transversions against genetic divergence using DAMBE (*Xia, 2013*).

A phylogenetic tree was constructed using the ML method with IQ-TREE (*Nguyen et al., 2015*; *Zhang et al., 2019*). The best-fit model was selected for each partition under corrected AIC using Partition Finder 2 (Table S3) (*Lanfear et al., 2017*) and evaluated using the ultrafast bootstrap approximation approach for 10,000 replicates. BI was performed using MrBayes version 3.2.6 (*Suchard & Huelsenbeck, 2012*; *Zhang et al., 2019*). Following the partition schemes suggested by Partition Finder, all model parameters were set as unlinked across partitions. Two independent runs with four simultaneous Markov chains (one cold and three incrementally heated at $T = 0.2$) were performed for 100 million generations, with sampling every 1,000 generations.

## RESULTS

### Mitogenomic characteristics of *Olidiana* species

The complete mitogenomes of five *Olidiana* species, namely, *O. alata* (MN780581; length 15,205 bp), *O. longsticka* (MN780582; length 15,993 bp), *O. olbliquea* (MN780583; length 15,312 bp), *O. ritcheri* (MN780584; length 15,372 bp), and *O. tongmaiensis* (MN780585; length 15,363 bp), were sequenced and assembled (Table S2). Their lengths were within the ranges of complete mitogenomes reported for other Cicadellidae species (14,805 bp for *Nephotettix cincticeps* and 16,811 bp for *Idioscopus laurifoliae*) (*Song, Cai & Li, 2017*; *Wang et al., 2018*). The mitogenomic architecture closely matched that of the inferred insect ancestral mitogenome (*Crease, 1999*): the newly sequenced mitogenomes had closed, circular DNA, typically comprising 37 genes (13 PCGs, 22 tRNAs, and 2 rRNAs) and a noncoding control region (Fig. 1). Of the 37 genes, most were encoded by the majority strand (J-strand) (9 PCGs and 14 tRNAs), whereas the minority strand (N-strand) encoded 14 genes (4 PCGs, 2 rRNAs, and 8 tRNAs) (Fig. 1, Table S4). However, the lengths of the 37 genes did not significantly differ between the five *Olidiana* species and other Cicadellidae species. The AT content of the five mitogenomes ranged from 78.0% (for *O. alata*) to 79.7% (for *O. longsticka*) and displayed a positive AT skew [0.147 (for *O. longsticka*) to 0.195 (for *O. tongmaiensis*)] and a negative GC skew [ −0.269 (for *O. tongmaiensis*) to −0.202 (for *O. longsticka*)] (Table 1). Additionally, the five *Olidiana* mitogenomes comprised 1–4-bp-long intergenic spacers at eight different loci, except for the *trnY–COI* intergenic spacer, which had 2–10-bp-long intergenic spacers. A total of 12 gene pairs were directly adjacent to each other, whereas the other gene pairs overlapped with each other, with overlap lengths of 1–4 bp, except for *trnW–trnC* and *trnS2–ND1*, which had large overlap lengths of 7–15 bp (Table S4). The gaps among the 37 genes in the mitogenomes were relatively smaller than those among genes in most reported Cicadellidae mitogenomes (*Wang et al., 2018*; *Wang et al., 2019a*; *Wang et al., 2019b*; *Wang et al., 2019c*; *Wang et al., 2020a*; *Wang et al., 2020b*).
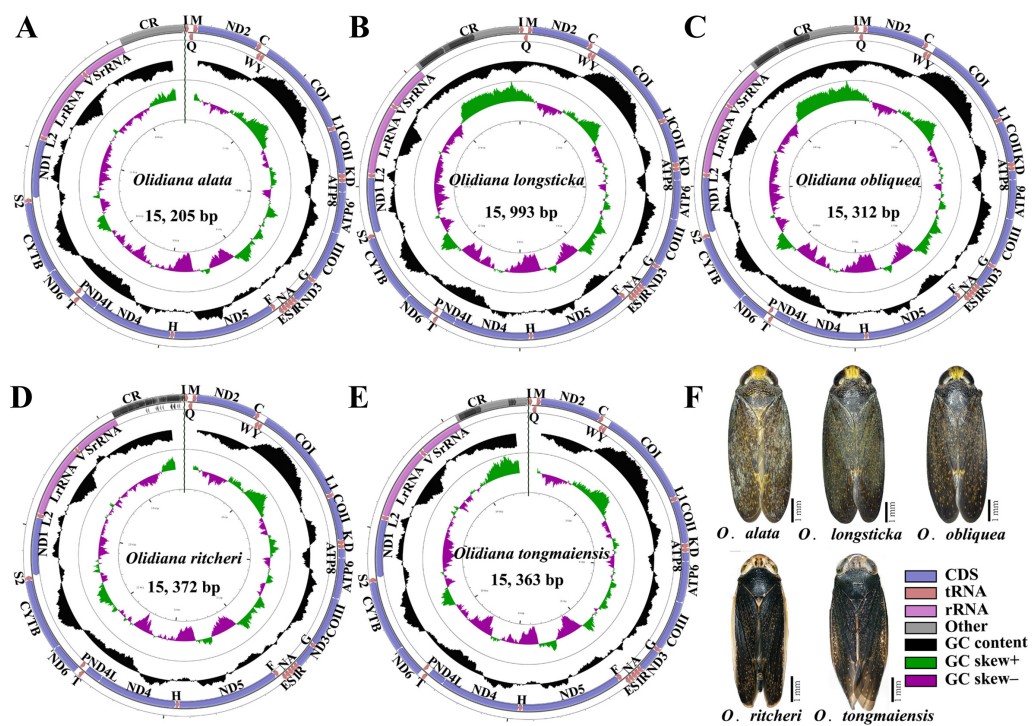

**Figure 1** **Mitochondrial genome map of five representative *Olidiana* species.** (A) Mitochondrial genome map of *Olidiana alata*. (B) Mitochondrial genome map of *Olidiana longsticka*. (C) Mitochondrial genome map of *Olidiana olbliquea*. (D) Mitochondrial genome map of *Olidiana ritcheri*. (E) Mitochondrial genome map of *Olidiana tongmaiensis*. (F) Dorsal view of five representative *Olidiana* species.

## PCGs and codon usage

Similar to that in other reported leafhopper mitogenomes (*Wang et al., 2019b*; *Wang et al., 2019c*; *Wang et al., 2020a*; *Wu et al., 2016*; *Yang, Mao & Bennett, 2017*), in the five *Olidiana* mitogenomes, the lengths of the 13 PCGs ranged from 150 bp (*ATP8*) to 1,674 bp (*ND5*) (Table S4). The AT content of the 13 PCGs ranged from 76.6% to 78.5%. Furthermore, the PCGs displayed positive AT skew (0.157–0.214) and negative GC skew (−0.299 to −0.234) (Table 1). Four PCGs (*ND4*, *ND4L*, *ND5*, and *ND1*) were coded by the N-strand, whereas the other nine (*COI*, *COII*, *COIII*, *ATP8*, *ATP6*, *ND2*, *ND3*, *ND6*, and *CYTB*) were coded by the J-strand. The *Olidiana* mitogenomes contained similar start and stop codons, and most PCGs had the typical start codon ATN (ATA/ATT/ATG/ATC) and either TAR (TAA/TAG) or an incomplete (single T) stop codon (Table S4). The presence of incomplete stop codons is a common feature of the mitochondrial genes among other leafhoppers, particularly of *ATP8*, and these incomplete stop codons are most likely caused by post-transcriptional modifications during mRNA maturation (*Wang et al., 2018*; *Wang et al., 2019a*; *Wang et al., 2019b*; *Wang et al., 2019c*; *Wang et al., 2020a*; *Wang et al., 2020b*; *Yuan et al., 2019*).

To understand the codon bias of the newly sequenced mitogenomes, RSCU and codon usage were determined. Codon usage was considerably similar between the five *Olidiana*

**Table 1  Nucleotide composition and skewness of five *Olidiana* mitogenomes.**

| Regions | Species | Length (bp) | AT (%) | GC (%) | AT skew | GC skew |
|---|---|---|---|---|---|---|
| Whole genome | O. alata | 15,205 | 78.0 | 22.0 | 0.169 | −0.245 |
| | O. longsticka | 15,993 | 79.7 | 20.3 | 0.147 | −0.202 |
| | O. olbliquea | 15,312 | 79.3 | 20.7 | 0.155 | −0.227 |
| | O. ritcheri | 15,372 | 78.2 | 21.8 | 0.151 | −0.257 |
| | O. tongmaiensis | 15,363 | 78.1 | 21.9 | 0.195 | −0.269 |
| 22 tRNAs | O. alata | 1,411 | 79.4 | 20.6 | 0.118 | −0.126 |
| | O. longsticka | 1,410 | 79.3 | 20.7 | 0.111 | −0.130 |
| | O. olbliquea | 1,400 | 79.6 | 20.4 | 0.111 | −0.110 |
| | O. ritcheri | 1,400 | 78.9 | 21.1 | 0.118 | −0.147 |
| | O. tongmaiensis | 1,411 | 79.3 | 20.7 | 0.145 | −0.150 |
| 13 PCGs | O. alata | 10,998 | 78.5 | 23.3 | 0.157 | −0.253 |
| | O. longsticka | 10,890 | 76.6 | 21.4 | 0.158 | −0.234 |
| | O. olbliquea | 10,890 | 78.2 | 21.8 | 0.166 | −0.239 |
| | O. ritcheri | 10,887 | 77.0 | 23.0 | 0.169 | −0.261 |
| | O. tongmaiensis | 10,886 | 76.6 | 23.4 | 0.214 | −0.299 |
| 2 rRNAs | O. alata | 1,915 | 80.7 | 19.3 | 0.199 | −0.306 |
| | O. longsticka | 1,915 | 82.1 | 17.9 | 0.189 | −0.307 |
| | O. olbliquea | 1,968 | 81.4 | 18.6 | 0.199 | −0.290 |
| | O. ritcheri | 1,955 | 81.3 | 18.7 | 0.171 | −0.294 |
| | O. tongmaiensis | 1,911 | 81.0 | 19.0 | 0.231 | −0.284 |
| Control region | O. alata | 1,017 | 85.3 | 14.7 | 0.048 | −0.061 |
| | O. longsticka | 1,804 | 84.1 | 15.9 | 0.065 | 0.057 |
| | O. olbliquea | 1,075 | 85.8 | 14.2 | 0.042 | −0.085 |
| | O. ritcheri | 1,149 | 84.4 | 15.6 | 0.012 | −0.231 |
| | O. tongmaiensis | 1,164 | 85.3 | 14.7 | 0.048 | −0.061 |

mitogenomes and other Cicadellidae mitogenomes (Fig. 2). Among the five *Olidiana* mitogenomes, the most frequently used codon was UUA (for leucine; Leu). Leucine (Leu) 300–347, isoleucine (Ile) 350–404, methionine (Met) 353–375, and phenylalanine (Phe) 282–295 were the most frequently coded amino acids. However, five codons, including UCA, ACC, GUG, CCG, and GCG, were seldom used (Fig. 2). The codon usage pattern of Coelidiinae mitogenomes is highly consistent with that of previously sequenced Cicadellidae mitogenomes (*Du, Dai & Dietrich, 2017a*; *Wang et al., 2020a*; *Wang et al., 2020b*).

## tRNAs and rRNAs

Consistent with most reported leafhopper mitogenomes (*Li et al., 2017b*; *Wang, Li & Dai, 2017*; *Wang et al., 2018*; *Wang et al., 2019b*; *Wang et al., 2019c*; *Wang et al., 2020a*; *Wang et al., 2020b*), the five *Olidiana* mitogenomes contained 22 tRNA genes, ranging from 57 bp (for *trnC*; *O. longsticka*) to 73 bp (for *trnK*; *O. tongmaiensis*) in length. The AT content of the tRNA genes ranged from 78.9% to 79.4%, displaying a positive AT skew (0.111–0.145) and negative GC skew (−0.150 to −0.110) (Table 1). In addition, all tRNAs exhibited a highly conserved, canonical cloverleaf secondary structure, except trnS1 (AGN), which lacked the stable dihydrouridine arm commonly found in most hemipterans (Fig. 3) (*Cameron,*

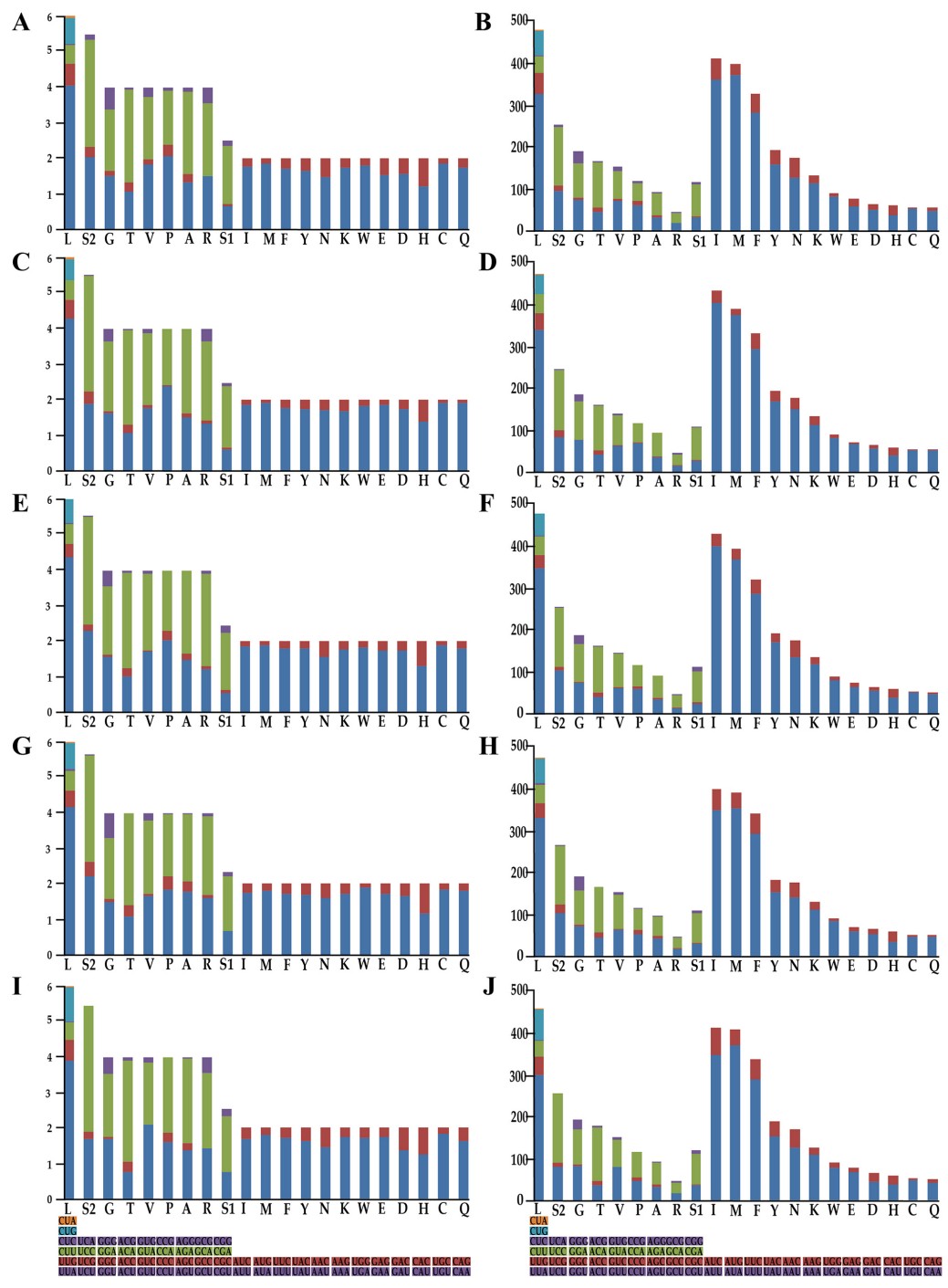

**Figure 2  Relative synonymous codon usage and number of codons used in *Olidiana* mitogenomes.**
(A) Relative synonymous codon usage in the *O. alata* PCGs of the mitogenome. (B) Number of codon used in *O. alata* PCGs of the mitogenome. (C) Relative synonymous codon usage in the *O. longsticka* PCGs of the mitogenome. (D) Number of codon used in *O. longsticka* (continued on next page...)

**Figure 2 (...continued)**
PCGs of the mitogenome. (E) Relative synonymous codon usage in the *O. olbliquea* PCGs of the mitogenome. (F) Number of codon usage in *O. olbliquea* PCGs of the mitogenome. (G) Relative synonymous codon usage in the *O. ritcheri* PCGs of the mitogenome. (H) Number of codon used in *O. ritcheri* PCGs of the mitogenome. (I) Relative synonymous codon usage in the *O. tongmaiensis* PCGs of the mitogenome. (J) Number of codon used in *O. tongmaiensis* PCGs of the mitogenome.

*2014*; *Li et al., 2017b*; *Wang, Li & Dai, 2017*; *Wang et al., 2018*; *Wang et al., 2019b*; *Wang et al., 2019c*; *Wang et al., 2020a*; *Wang et al., 2020b*).

Two rRNA genes (*rrnL* and *rrnS*) are highly conserved in Cicadellidae mitogenomes, and each of the five *Olidiana* mitogenomes contained these two rRNA genes. *rrnL* ranged from 1,176 bp (for *O. ritcheri*) to 1,186 bp (for *O. longsticka*) in length, whereas *rrnS* ranged from 729 bp (in *O. alata*) to 788 bp (in *O. obliquea*) in length (Fig. 1, Table 1). The rRNA genes of *Olidiana* mitogenomes displayed a positive AT skew (0.171–0.231) and negative GC skew (−0.307 to −0.284) (Table 1). *rrnL* was located between *trnL2* and *trnV*, and *rrnS* was located between *trnV* and the control region (Table S4).

## Control region

The control region of *Olidiana* mitogenomes ranged from 1,075 bp (for *O. alata*) to 1,804 bp (for *O. longsticka*). The differences in length in the control region were mainly attributed to the length and number of tandem repeats (R). All variable repeats in *Olidiana* mitogenomes were identified. Only a short unit (R) with two copies, both 115 bp in length, was present in *O. alata*. In *O. longsticka* and *O. olbliquea*, the first repeat region (R1) was 448 and 226 bp in length, respectively, and both comprised two units. The other two repeat regions, i.e., R2 and R3, were located after R1, and they were 345 and 433 bp (*O. longsticka*) and 129 and 28 bp (*O. olbliquea*) in length, respectively; both comprised three copies. *O. ritcheri* and *O. tongmaiensis* comprised four types of units: R1, 2 × 135 bp; R2, 2 × 195 bp; R3, 3 × 116 bp; and R4, 4 × 78 bp and R1, 2 × 281 bp; R2, 3 × 191 bp; R3, 3 × 81 bp; and R4, 3 × 4 bp, respectively (Fig. 4). Similar to the long intergenic spacers in other insect species, the repeat regions in leafhoppers may be attributed to an alternative origin of mitogenome replication (*Dotson & Beard, 2001*). The AT content (84.1%–85.8%) of the control regions was generally higher than that of the other regions. This is in part due to damage or accumulation of mutations in the mitochondrial DNA (*Martin, 1995*). The control regions of the five *Olidiana* mitogenomes displayed a slightly positive AT skew [ranging from 0.012 (for *O. ritcheri*) to 0.065 (for *O. longsticka*)] and negative GC skew [ranging from −0.231 (for *O. ritcheri*) to −0.061 (for *O. alata*)], except the control region of *O. longsticka*, which displayed a slightly positive GC skew (0.057) (Table 1). Moreover, these control regions were compared with previously reported control region sequences; their differences were very large, and no obvious correlation or similarity was found with existing sequences.

## Phylogenetic relationships

No saturation was detected among the four candidate nucleotide sequence datasets (PCGs, PCG12, PCG12R, and PCGR) prepared for ML and BI analyses (all *Iss <Iss.cSym* or

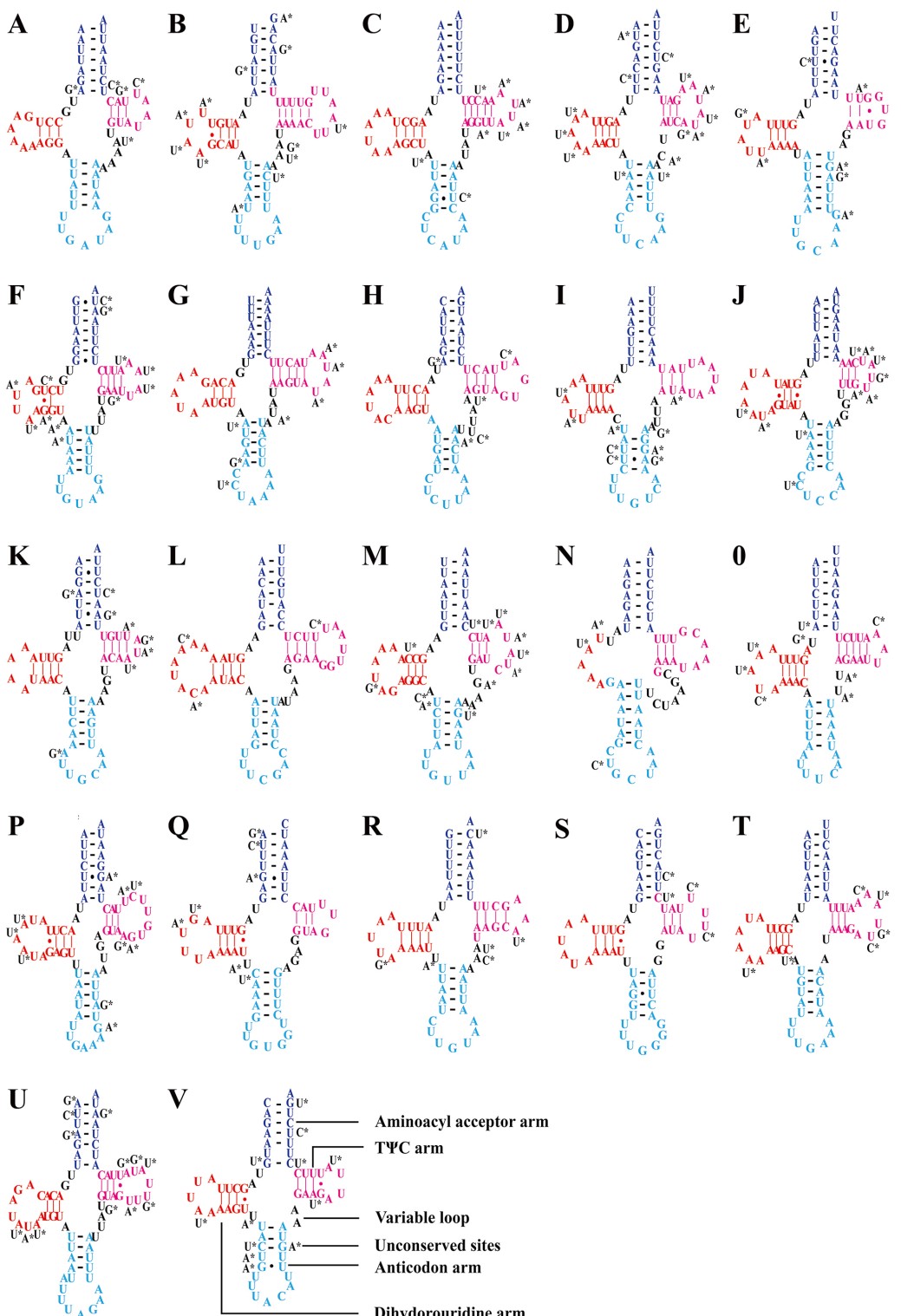

**Figure 3** **Predicted secondary structures of tRNAs in of the five *Olidiana* species.** (A) Predicted secondary structures of Isoleucine (Ile, I). (B) Predicted secondary (continued on next page...)

**Figure 3 (...continued)**
structures of Glutamine (Glu, Q). (C) Predicted secondary structures of Methioine(Met, M). (D) Predicted secondary structures of Tryptophan (Trp, W). (E) Predicted secondary structures of Cysteine (Cys, C). (F) Predicted secondary structures of Tyrosine (Tyr, Y). (G) Predicted secondary structures of Leucine (Leu , L1). (H) Predicted secondary structures of Lysine (Lys, K). (I) Predicted secondary structures of Aspartic (Asp, D). (J) Predicted secondary structures of Glycine (Gly, G). (K) Predicted secondary structures of Tyrosine (Tyr, Y). (L) Predicted secondary structures of Arginine (Arg, R). (M) Predicted secondary structures of Asparagine (Asn, N). (N) Predicted secondary structures of Serine (Ser, S1). (O) Predicted secondary structures of Glutamic (Glu, E). (P) Predicted secondary structures of Phenylalanine (Phe, F). (Q) Predicted secondary structures of Histidine (His, H). (R) Predicted secondary structures of Threonine (Thr, T). (S) Predicted secondary structures of Proline (Pro, P). (T) Predicted secondary structures of Serine (Ser, S2). (U) Predicted secondary structures of Leucine (Leu, L2). (V) Predicted secondary structures of Valine (Val, V). Dashes (–) indicate Watson–Crick base pairing and dots (●) indicate G–U base pairing.

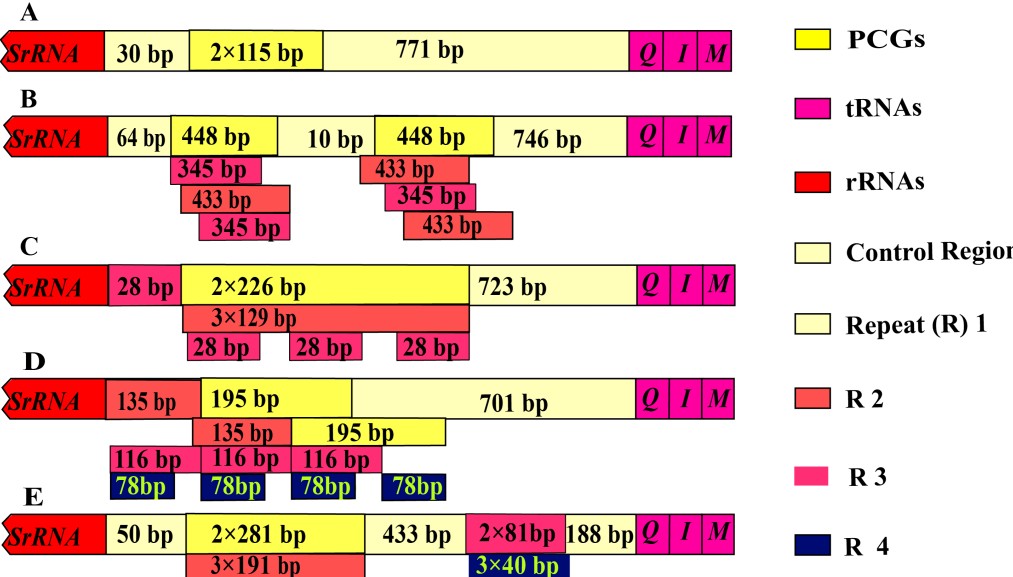

**Figure 4** **Organization of the control region in the complete mitogenome of five *Olidiana* species.** (A) Organization of the control region in the complete mitogenome of *Olidiana alata*. (B) Organization of the control region in the complete mitogenome of *Olidiana longsticka*. (C) Organization of the control region in the complete mitogenome of *Olidiana olbliquea*. (D) Organization of the control region in the complete mitogenome of *Olidiana ritcheri*. (E) Organization of the control region in the complete mitogenome of *Olidiana tongmaiensis*.

*Iss. cAsym*; $P < 0.05$) (Table S5), and the concatenated data were deemed suitable for phylogenetic analysis. Therefore, 10 phylogenetic trees were reconstructed based on five datasets (PCGs, PCG12, AA, PCG12R, and PCGR) using the BI and ML methods, and the main topological structures of the constructed phylogenetic trees were consistent (Figs. 5 and 6). Our results support the view that treehoppers originated from Cicadellidae and further confirm that Cicadellidae is a paraphyletic group, which is also supported by many previous studies (*Du, Dai & Dietrich, 2017a*; *Du, Dietrich & Dai, 2019*; *Hu et al., 2019*; *Mao, Yang & Bennett, 2016*; *Song, Cai & Li, 2017*; *Wang et al., 2018*; *Wang et al., 2020a*; *Wang et al., 2020b*; *Yu et al., 2017*). All analyses clearly support the monophyletic relationship of

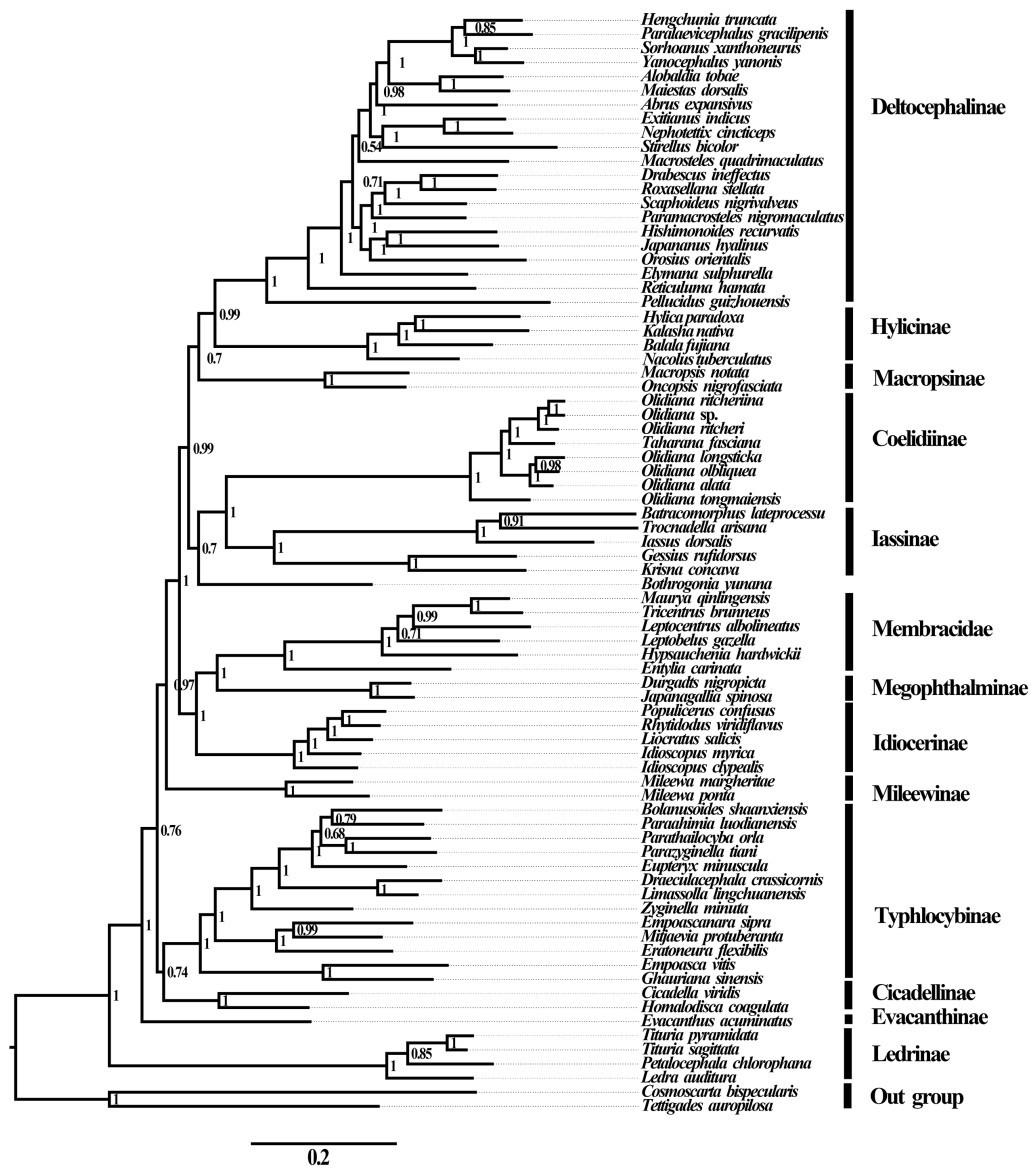

**Figure 5** Phylogenetic trees of Cicadellidae inferred by MrBayes 3.2.6 based on nucleotides of the first and second codons of 13 PCGs and two rRNAs (BI-13PCG12-2R).

the 10 subfamilies within Cicadellidae and confirm that Iassinae and Coelidiinae as well as Megophthalminae and treehoppers are sister groups (Figs. 5 and 6, Figs. S1–S7). These results are consistent with those of previous studies (*Wang, Li & Dai, 2017*; *Wang et al., 2018*; *Wang et al., 2019b*; *Wang et al., 2019c*; *Wang et al., 2020a*; *Wang et al., 2020b*; *Wu et al., 2016*; *Yang, Mao & Bennett, 2017*).

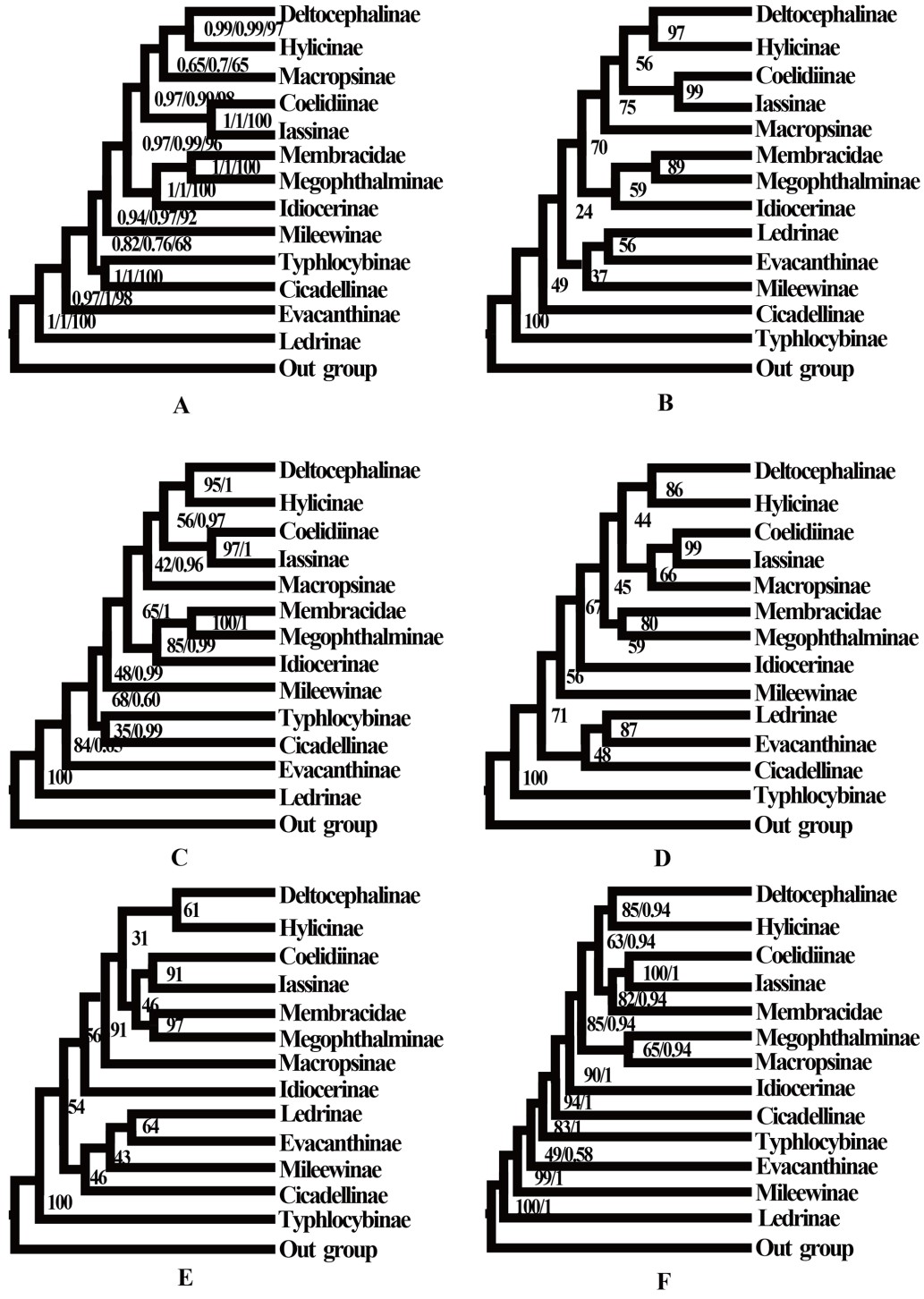

**Figure 6** **Phylogenetic trees of leafhoppers.** (A) Phylogenetic trees of leafhoppers inferred by the Mr-Bayes 3.2.6 and maximum likelihood methods (continued on next page...)

**Figure 6 (…continued)**
(BI-13PCG12/BI-13PCG12-2R/ML-13PCG-2R). (B) Phylogenetic trees of leafhoppers inferred by the maximum likelihood method based on nucleotide sequences from the first and second codons of 13 PCGs (ML-PCG12). (C) Phylogenetic trees of leafhoppers inferred by the MrBayes 3.2.6 based on nucleotides of the first and second codons of 13 PCGs and 2 rRNAs (BI-PCG12-2R); and maximum likelihood 13 PCGs and 2 rRNAs (ML-13PCG-2R). (D) Phylogenetic trees of leafhoppers inferred by the maximum likelihood 13 PCGs (ML-13PCG). (E) Phylogenetic trees of leafhoppers inferred by the maximum likelihood 13 PCGs and 2 rRNAs (ML-13PCG-2R). (F) Phylogenetic trees of leafhoppers inferred by the maximum likelihood and MrBayes 3.2.6 methods based on amino acid sequences of 13 PCGs (ML/BI-AA).

## DISCUSSION

The phylogenetic positions of the subfamilies Ledrinae and Deltocephalinae were different from those observed in previous studies, in which Deltocephalinae was located at the base of the Cicadellidae phylogenetic tree (*Du et al., 2017b*; *Wang et al., 2019b*; *Wang et al., 2019c*). However, in our study, the main topological structure showed that Ledrinae, instead of Deltocephalinae, was located at the base of the phylogenetic tree. This result confirms that Ledrinae is an ancient group of leafhoppers. This result is in agreement with that reported by *Chen et al. (2019)*, *Wang et al. (2019a)* and *Wang et al. (2019b)* and differs from that reported by *Du, Dai & Dietrich (2017a)*, *Du et al. (2017b)* and *Tang, Huang & Zhang (2020)*. Moreover, *Tang, Huang & Zhang (2020)* reported that there is a stable relationship between the subfamilies [(Coelidiinae + Iassinae) + Hylicinae]. The results of the present study do not show such phylogenetic relationships but rather indicate that Deltocephalinae and Hylicinae are sister groups (Figs. 5 and 6, Figs. S1–S7). The phylogenetic results are different from those of similar recent studies, for which there are two possible reasons: (1) we included all 13 PCGs and 2 rRNA genes in the phylogenetic analysis, whereas previous studies included 13 PCGs or selected a small number of species; (2) we included different species in the phylogenetic analysis, resulting in different conserved blocks after completion of multiple sequence alignment. In addition, phylogenetic results based on mitogenomes showed that Iassinae and Coelidiinae are sister groups; however, the phylogenetic results are different from those of *Dietrich et al. (2017)*. Based on anchored-hybrid enrichment data, the phylogenetic results showed that the main topology was as follows: (Coelidiinae + (Ledrinae + (Hylicinae + Neobalinae))). This difference can be mainly attributed to the phylogenetic results based on different molecular data and taxa. Additional data and taxon sampling of Evacanthinae, Hylicinae, Ledrinae, Neobalinae, and Macropsinae are required to reliably determine the relationship among Cicadellidae species.

Within Coelidiinae, all phylogenetic relationships demonstrated high nodal support in both ML analyses. Interestingly, instead of clustering together, the seven *Olidiana* species were further divided into three groups in all analyses. The first group comprised three *Olidiana* species (*O. ritcheriina* + (*Olidiana* sp. + *O. ritcheri*)), which were sister groups to *H. fascianus* (= *T. fasciana*, NC036015). The second group comprised three *Olidiana* species (*O. longsticka* + (*O. olbliquea* + *O. alata*)), whereas the third group comprised only *O. tongmaiensis*. These results indicate the complex phylogenetic relationship among the species in *Olidiana* genus and its related genera. In particular, *O. tongmaiensis*, which is only distributed in the Palearctic realm, diverged from the other *Olidiana* species

(distributed in the Oriental realm; this finding is consistent with their biogeographic patterns) (*Li & Fan, 2017*; *Viraktamath & Meshram, 2019*; *Zhang, 1990*). Interestingly, the seven *Olidiana* species could be divided into three groups based on significant differences in their morphological characteristics, which were characterized by body color, shape, and position of the processes on the aedeagus shaft. Therefore, based on complete mitogenome phylogenetic analysis and comparison of morphological characteristics, we propose *Olidiana* as a paraphyletic genus and suggest that it should be further examined based on the shape and position of the processes on the aedeagus shaft.

## CONCLUSIONS

In this study, we sequenced and annotated the complete mitogenomes of five *Olidiana* species. The general genomic characteristics (gene content, gene size, gene order, base composition, PCG codon usage, and tRNA secondary structure) of the *Olidiana* mitogenomes were mostly consistent with those of reported Cicadellidae mitogenomes. In addition, we performed phylogenetic analyses to infer the probable relationships among the Cicadellidae subfamilies as well as to confirm the phylogenetic relationship among the *Olidiana* species. Our results support the presence of a monophyletic relationship among the 10 Cicadellidae subfamilies and confirm that Iassinae and Coelidiinae are sister groups with high approval ratings. Interestingly, phylogenetic analyses of the mitogenomes support our assertion that *Olidiana* is a paraphyletic genus, with the following topology: (*O. tongmaiensis* + (*O. longsticka* + (*O. olbliquea* + *O. alata*)) + (*H. fascianus* + (*O. ritcheriina* + (*Olidiana* sp. + *O. ritcheri*)))). Our findings will not only improve our understanding of the phylogenetic relationships of related insects but also contribute toward their taxonomic classification within Cicadellidae. Further studies of the combination of morphological and molecular characteristics of additional species are warranted to confirm the taxonomy of Cicadellidae.

### Funding

This work was supported by the National Natural Science Foundation, China [Grant no. 31672342] and the Program of Excellent Innovation Talents, Guizhou Province, China [Grant no. 20164022]. The funders had no role in study design, data collection and analysis, decision to publish, or preparation of the manuscript.

### Grant Disclosures

The following grant information was disclosed by the authors:
National Natural Science Foundation, China: 31672342.
Program of Excellent Innovation Talents, Guizhou Province, China: 20164022.

### Competing Interests

The authors declare there are no competing interests.

## Author Contributions

- Xianyi Wang conceived and designed the experiments, performed the experiments, analyzed the data, prepared figures and/or tables, authored or reviewed drafts of the paper, and approved the final draft.
- Jiajia Wang conceived and designed the experiments, performed the experiments, prepared figures and/or tables, and approved the final draft.
- Ren-Huai Dai conceived and designed the experiments, authored or reviewed drafts of the paper, and approved the final draft.

## Field Study Permissions

The following information was supplied relating to field study approvals (i.e., approving body and any reference numbers):

The Forestry and Grassland Administration of Yunnan Province approved the collection of leafhopper specimens in Yunnan Province (NO. 2019.209).

## DNA Deposition

The following information was supplied regarding the deposition of DNA sequences:

The datasets analyzed for this study are available at NCBI GenBank: MN780581–MN780585.

Data is also available in the Supplemental Files.

## Data Availability

The datasets analyzed for this study are available at GenBank: *O. alata* (MN780581), *O. longsticka* (MN780582), *O. olbliquea* (MN780583), *O. ritcheri* (MN780584), *O. tongmaiensis* (MN780585).

## Supplemental Information

Supplemental information for this article can be found online at http://dx.doi.org/10.7717/peerj.11086#supplemental-information.

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
