# Peer review of "Mitogenomics of five Olidiana leafhoppers (Hemiptera: Cicadellidae: Coelidiinae) and their phylogenetic implications"

_PeerJ, doi:10.7717/peerj.11086_

## Round 0.1 · original submission · Major Revisions

Dear Drs. Wang and colleagues:

Thanks for submitting your manuscript to PeerJ. I have now received three independent reviews of your work, and as you will see, one reviewer recommended rejection, while another suggested a major revision. I am affording you the option of revising your manuscript according to all three reviews but understand that your resubmission may be sent to at least one new reviewer for a fresh assessment (unless the reviewer recommending rejection is willing to re-review).

In general, the reviewers wish to see more comparisons with other related genomes. The distinctive features of these novel genomes are not made clear enough. Please address this to increase the relevance of your study.

Importantly, please ensure that an English expert has edited your revised manuscript for content and clarity. Please also ensure that your figures and tables contain all of the information that is necessary to support your findings and observations, including complete legends.

Thus, I encourage you to revise your manuscript, accordingly, taking into account all of the concerns raised by the three reviewers.

Good luck with your revision,

-joe

Reviewer 1 ·

Basic reporting

This study described the mitochondrial genomes of 5 species of the leafhopper genus Olidiana. It continues a series of recent papers that sequenced leafhopper mtgenomes, provided a standard set of genome composition statistics and comparisons to other leafhopper mtgenomes, and combined the data with those generated by previous analyses to produce a phylogenetic estimate of Cicadellidae. Given the recent flood of new mtgenomes for leafhoppers, the results of this study are only marginally novel because previous studies have included representatives of the subfamily Coelidiinae, including Olidiana species, and there does not appear to be anything particularly new or unusual about the newly described genomes. The phylogenetic results are somewhat different from those of similar recent studies and this may be a result of adding data for additional species of Coelidiinae but the authors do not attempt to account for the differences between their results and those of other recent mitogenome-based phylogenies of Cicadellidae. The following issues need to be addressed:

1. In the introduction it is stated that Olidiana is the largest group of Coelidiinae. This is not true. Thagria has many more species. Also, Olidiana species have not been reported “worldwide”; they occur only in the Oriental region.
2. The Results section includes many details on various characteristics of the sequenced mitogenomes but the authors should add some comments to the Discussion explaining whether they found these genomes to include any unusual features compared to known genomes for other Coelidiinae and for Cicadellidae in genera. For example, were there any gene rearrangements?
3. The caption for Figure 5 should explain what the numbers on branches of the tree represent. Also, the second node from the base of the Coelidiinae clade has no support values indicated. Does this mean that this node received <50% support?
4. The whole manuscript needs to be edited to improve the English.

Experimental design

no comment

Validity of the findings

The findings appear to be reported in adequate detail but a more concise statement explaining whether these new genomes contain any unusual features not observed previously in cicadellid genomes needs to be added.

Reviewer 2 ·

Basic reporting

The English language should be improved. It should be checked by a native English speaker.
Background of the manuscript is a bit hazy, and a number of what appear to be relevant papers were not referenced in the manuscript. The authors should rework the paper and include a broader literature review.

Experimental design

There are few distinct research questions asked or answered.

Validity of the findings

There is no particular novelty in the manuscript. It just reports the new mitogenomes of five Olidiana species with the basic stats.

Additional comments

The MS presents the mitochondrial genomes of five leafhoppers of the genus Olidiana from the subfamily Coelidiinae assembled using the NGS method. The methodology is basically good, but considering that both the subfamily Coelidiinae and the family Cicadellidae are very large and diverse groups the comparative mitogenomic analyses and phylogenetic analysis are relatively weak (see the Results and Discussion part of the manuscript), so in its current form the paper is not suitable for publication in this journal.

Reviewer 3 ·

Basic reporting

no comment

Experimental design

no comment

Validity of the findings

no comment

Additional comments

Wang et al sequenced five Olidiana mitogenoes and constructed phylogenetic relationship among subfamilies within Cicadellidae. Overall the manuscript was enriched with figures and tables, and the results could be useful in further Cicadellidae studies.
1. Many redundant informtion for genome assembly, annotation and analyses. Authors should reorganize the structures and delete repeated sentences.
2. I am not sure that the software Mega could be used for sequence concatenation.
3. More detailed information needs to be provided, such as bootstrap, how to determine the chain convergence.
4. Authors should provide detailed comparative mitogenomic analyses between the five Olidiana species, and discussed with other Coelidiinae species of previous studies. However, authors only provided some simple descriptions for these mitogenomes. More discussion is needed.
5. How many topologies were obtained based on five datasets and two methods? How many topologies among subfamilies? Provide topology test results.
6. English language should be improved.

---

## Round 0.2 · Major Revisions

Dear Drs. Wang and colleagues:

Thanks for revising your manuscript. The lone reviewer is somehwhat satisfied with your revision; however, there remain several issues to address. Please address these ASAP and resubmit another revision. Also, please have an English expert help with the writing of your revision.

Best,

-joe

Reviewer 1 ·

Basic reporting

This version of the manuscript has been improved substantially but there are still some problems with the English. The authors should consult a colleague who is a native English speaker to fix several problems with the narrative.

Experimental design

Good.

Validity of the findings

As pointed out in a previous review, there have been several mtgenome-based phylogenies of leafhoppers published recently. The main novelty of this paper is its inclusion of multiple species of one genus, which allowed the authors to test the monophyly of the genus. The finding that Olidiana is probably paraphyletic is significant.

Additional comments

Another recent paper (Tang et al. 2020, Insects) added the first representatives of Hylicinae, showing that this group is closely related to Coelidiinae and Iassinae. The authors may want to add the data for Hylicinae to their dataset and re-do their analysis. Also, it may be a good idea to compare the phylogenetic results with those of Dietrich et al. (2017, Insect Systematics and Diversity) based on anchored-hybrid enrichment data. The latter analysis recovered Coelidiinae (excluding Equeefini) as a monophyletic group and also resolved relationships among tribes with strong support but the relationship of Coelidiinae to other leafhopper subfamilies was poorly resolved.

---

## Round 0.3 · accepted · Accept

Dear Drs. Wang and colleagues:

Thanks for revising your manuscript based on the concerns raised by the reviewers. I now believe that your manuscript is suitable for publication. Congratulations! I look forward to seeing this work in print, and I anticipate it being an important resource for groups studying cicadellid systematics. Thanks again for choosing PeerJ to publish such important work.

Best,

-joe

Reviewer 1 ·

Basic reporting

This is the third version of the manuscript I have seen. The authors have substantially addressed my previous concerns so I think this version is acceptable for publication.

Experimental design

n/a

Validity of the findings

n/a

Additional comments

n/a